# Experimental Study on the Fracture Process Zone Characteristics in Concrete Utilizing DIC and AE Methods

**Shuhong Dai [1],\*, Xiaoli Liu [2],\*** 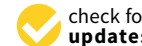 **and Kumar Nawnit [2]**

1 School of Mechanics and Engineering, Liaoning Technical University, Fuxin 123000, China
2 State Key Laboratory of Hydroscience and Engineering, Tsinghua University, Beijing 100084, China; nnet55@gmail.com
\* Correspondence: dsh3000@126.com (S.D.); xiaoli.liu@tsinghua.edu.cn (X.L.)

**Abstract:** The present work focuses on investigating the characteristics of the fracture process zone (FPZ) in concrete. The Single-edge notched (SEN) concrete beams under three-points bending are employed for conducting mode I fracture propagation. The displacement fields on the specimen surface and the internal AE signal of specimen are obtained simultaneously in real time by digital image correlation (DIC) and acoustic emission (AE) techniques. The experimental and analytical results indicated that the crack tip position, the crack extension length and the stress intensity factors (SIF) are obtained dynamically and quantitatively by DIC technique, and the length of FPZ is identified, respectively, by DIC and AE techniques in the crack extension process. The distribution of internal AE events is consistent with that of FPZ identified from surface deformation of specimens.

**Keywords:** fracture process zone; digital image correlation technique; acoustic emission technique; stress intensity factor

---

## 1. Introduction

Concrete is a quasi-brittle material. The micro-cracking region ahead of the real crack tip in concrete is defined as the fracture process zone (FPZ), and the characteristics of concrete FPZ have been a subject of massive experimental debates. Various techniques have been developed to track the evolution of micro-cracks in FPZ, such as the fiber optics technique [1], X-rays technique [2], fiber optic sensor [3], and optical microscopy [4]. Compared to the methods mentioned above, digital image correlation and acoustic emission techniques are increasingly popular recently.

The Digital image correlation (DIC) technique was firstly proposed by Peters and Sutton et al. [5,6]. It is an optical technique to determine surface deformations by matching the digital speckle images before and after deformation. DIC is a non-destructive and non-contact full field displacement measuring technique. The displacement and strain data obtained by DIC can be directly employed to theoretical analysis. Benefit from such advantages, the DIC has been widely applied to the study of concrete fracture. Choi and Shah [7] measured the lateral and axial deformations on a concrete specimen surface subjected to compression. Corr et al. [8] studied the interfacial transition zone between aggregates and cement paste in plain concrete. They also investigated its softening and fracture behavior. Wu et al. [9] studied the properties of the fracture process zone in concrete using DIC technique. Rouchier et al. [10] studied the whole process of concrete fracture using the DIC technique. The acoustic emission (AE) technique has been extensively applied in concrete engineering for approximately five decades since the 1960's [11]. The AE technique can continuously monitor the internal micro-cracks event and the failure process of concrete in real time, which makes it more

popular than other methods. Wells [12] studied AE waveforms and determined the relationship between strain measurements and AE events. A frequency analysis and a source location analysis were reported to demonstrate the relationships [13]. These studies have produced practical applications for monitoring micro-cracks in concrete structures and are very useful in diagnostic applications. AE technique was utilized to assess the concrete fracture process, as the pioneering research works of Colombo [14]. The research works of Colombo et al. [15] have indicated that concrete micro cracks emit waves possessed smaller amplitudes, whereas macro cracks emit waves possessed larger amplitudes. Muralidhara [16] shed lights on the relationship between the AE event and the evolution of the fracture process in concrete. Recently, the relationship between the formation of FPZ in concrete and AE events has been extensively studied [17–20]. In order to cover their weaknesses and improve their advantages, DIC and AE techniques were combined to monitor the concrete crack evolution process [21–23].

In spite of the extensive work and many successes in characterizing FPZ, the comprehension of the characteristics of concrete FPZ requires further research. In this presentation, the concrete fracture tests are conducted on Single-edge notched beams through three-points bending conditions. The DIC technique is applied to measure the displacement and strain fields around the crack tip in real-time. Then the SIF, crack tip position and crack length are quantitatively and dynamically derived from the displacement fields measured by DIC technique. The AE technique is applied to monitor the internal fracture events during the crack propagation process, in particular the subcritical extension process. In short, the objective of this research is to investigate the characteristics of FPZ during the concrete crack propagation process.

## 2. Experimental Procedure

### 2.1. Digital Image Correlation Technique

Digital image correlation method relies on observations of random speckle patterns on the specimen surface. Image patterns are recorded before and after deformation of the specimens, then they are digitized and stored in a computer. The undeformed and deformed images are divided into small regions called "subsets", with each subset containing a group of pixels. Digital image correlation is used to match the subsets on the undeformed image with their corresponding subsets on the deformed image, as shown in Figure 1.

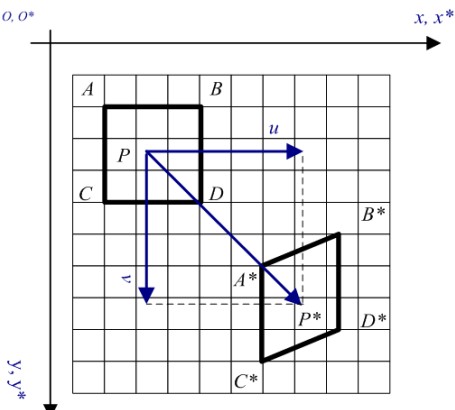

**Figure 1.** Initial subset (ABCD centered on *P*) and deformed subset (A\*B\*C\*D\* centered on *P*\*).

The cross-correlation function is chosen for this study, and it is defined as Equation (1):

$$F\left(x_m, y_n, u_x, u_y, \frac{\partial u_x}{\partial x}, \frac{\partial u_x}{\partial y}, \frac{\partial u_y}{\partial x}, \frac{\partial u_y}{\partial y}\right) = 1 - \frac{\sum\limits_{i=-\frac{R}{2}}^{\frac{R}{2}} \sum\limits_{j=-\frac{R}{2}}^{\frac{R}{2}} \left[\left(f(x_{m+i}, y_{n+j}) - \overline{f}\right) \cdot \left(g(x_{m+i}, y_{n+j}) - \overline{g}\right)\right]}{\left[\sum\limits_{i=-\frac{R}{2}}^{\frac{R}{2}} \sum\limits_{j=-\frac{R}{2}}^{\frac{R}{2}} \left(f(x_{m+i}, y_{n+j}) - \overline{f}\right)^2 \cdot \sum\limits_{i=-\frac{R}{2}}^{\frac{R}{2}} \sum\limits_{j=-\frac{R}{2}}^{\frac{R}{2}} \left(g(x_{m+i}, y_{n+j}) - \overline{g}\right)^2\right]^{\frac{1}{2}}} \tag{1}$$

The $f(x_i, y_j)$ represents the gray level value at coordinate $(x_i, y_j)$ of the undeformed image, while the $g(x_j^*, y_j^*)$ represents the gray level value at coordinate $(x_j^*, y_j^*)$ of the deformed image, the subset size is $N \times N$. The coordinates $(x_i, y_j)$ and $(x_j^*, y_j^*)$ are directly related by the deformation occurring between the two images. If the deformation occurs in two dimensions parallel to the camera, then the coordinates are related by Equation (2):

$$
\begin{aligned}
x^* &= x + u_x + \frac{\partial u_x}{\partial x}\Delta x + \frac{\partial u_x}{\partial y}\Delta y \\
y^* &= y + u_y + \frac{\partial u_y}{\partial x}\Delta x + \frac{\partial u_y}{\partial y}\Delta y
\end{aligned}
\tag{2}
$$

The $u_x$ and $u_y$ are the displacements of the subset center in the $x$ and $y$ directions, respectively, and $\Delta x$ and $\Delta y$ are distances from the subset center to any point in the subset $(x, y)$.

## 2.2. Estimation of Stress Intensity Factors

The method for estimating mode I and mixed-mode I-II stress intensity factors is proposed here, based on the displacement fields determined via the DIC method. The displacement fields around a crack tip of the concrete specimen are expressed as Equation (3):

$$
\begin{Bmatrix} u_x \\ u_y \end{Bmatrix} = \sum_{n=1}^{\infty} \frac{A_{In}}{2G} r^{n/2} \begin{Bmatrix} \kappa\cos\frac{n}{2}\theta - \frac{n}{2}\cos(\frac{n}{2}-2)\theta + \{\frac{n}{2}+(-1)^n\}\cos\frac{n}{2}\theta \\ \kappa\sin\frac{n}{2}\theta + \frac{n}{2}\sin(\frac{n}{2}-2)\theta - \{\frac{n}{2}+(-1)^n\}\sin\frac{n}{2}\theta \end{Bmatrix}
$$
$$
- \sum_{n=1}^{\infty} \frac{A_{IIn}}{2G} r^{n/2} \begin{Bmatrix} \kappa\sin\frac{n}{2}\theta - \frac{n}{2}\sin(\frac{n}{2}-2)\theta + \{\frac{n}{2}-(-1)^n\}\sin\frac{n}{2}\theta \\ -\kappa\cos\frac{n}{2}\theta - \frac{n}{2}\cos(\frac{n}{2}-2)\theta + \{\frac{n}{2}-(-1)^n\}\cos\frac{n}{2}\theta \end{Bmatrix}
\tag{3}
$$

The $u_x$ and $u_y$ are the displacement components in the specimen considered, and $G$ is shear modulus. The $\kappa$ is $(3-\mu)/(1+\mu)$ for plane stress, and $r$ and $\theta$ are the polar coordinates around a crack tip, as shown in Figure 2.

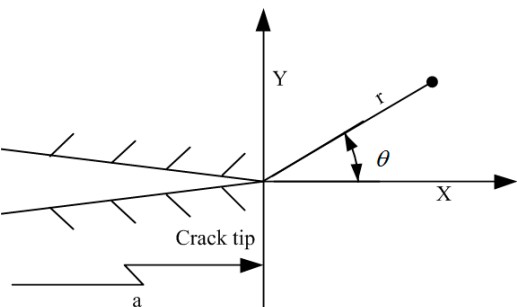

**Figure 2.** Coordinate system around a crack tip.

In the series solutions, the coefficient of the first terms $A_{I1}$ and $A_{II1}$ are related to stress intensity factors $K_I$ and $K_{II}$ of mode I and mode II through the relations of Equation (4).

$$
A_{I1} = \frac{K_I}{\sqrt{2\pi}}, A_{II1} = -\frac{K_{II}}{\sqrt{2\pi}}
\tag{4}
$$

For the cases of mode I fracture, the displacement component of the specimen is perpendicular to the crack surface. This means that the $u_y$ displacement field can be generally used to determine the stress intensity factors. In mixed-mode cases, however, the dominant displacement component for the crack can't be predicted before analysis. Therefore, radial $u_r$ and circumferential displacement components $u_\theta$ on a polar coordinate system are used in this study to transform Cartesian displacements $u_x$ and $u_y$. The displacements $u_r$ and $u_\theta$ are obtained as Equation (5):

$$\left\{ \begin{array}{c} u_r \\ u_\theta \end{array} \right\} = \left[ \begin{array}{cc} \cos\theta \, \sin\theta \\ -\sin\theta \, \cos\theta \end{array} \right] \left\{ \begin{array}{c} u_x \\ u_y \end{array} \right\} \tag{5}$$

The displacement fields in Equation (5) can, hence, be rewritten as Equation (6):

$$\left\{ \begin{array}{c} u_x \\ u_y \end{array} \right\} = \sum_{n=1}^{N} A_{\mathrm{I}n} \left\{ \begin{array}{c} f_{\mathrm{I}n}(r,\theta) \\ g_{\mathrm{I}n}(r,\theta) \end{array} \right\} - \sum_{n=1}^{N} A_{\mathrm{II}n} \left\{ \begin{array}{c} f_{\mathrm{II}n}(r,\theta) \\ g_{\mathrm{II}n}(r,\theta) \end{array} \right\} \tag{6}$$

The $N$ is the number of terms of the series expansion of the displacement field. From Equations (5) and (6) with rigid-body displacements taken into account, the displacement fields can be expressed as Equations (7) and (8):

$$\begin{aligned} u_{rk} = & \left\{ \sum_{n=1}^{N} A_{\mathrm{I}n} f_{\mathrm{I}n}(r_k,\theta_k) - \sum_{n=1}^{N} A_{\mathrm{II}n} f_{\mathrm{II}n}(r_k,\theta_k) \right\} \cos\theta_k \\ & + \left\{ \sum_{n=1}^{N} A_{\mathrm{I}n} g_{\mathrm{I}n}(r_k,\theta_k) - \sum_{n=1}^{N} A_{\mathrm{II}n} g_{\mathrm{II}n}(r_k,\theta_k) \right\} \sin\theta_k + T_x \cos\theta_k + T_y \sin\theta_k \end{aligned} \tag{7}$$

$$\begin{aligned} u_{\theta k} = & -\left\{ \sum_{n=1}^{N} A_{\mathrm{I}n} f_{\mathrm{I}n}(r_k,\theta_k) - \sum_{n=1}^{N} A_{\mathrm{II}n} f_{\mathrm{II}n}(r_k,\theta_k) \right\} \sin\theta_k \\ & + \left\{ \sum_{n=1}^{N} A_{\mathrm{I}n} g_{\mathrm{I}n}(r_k,\theta_k) - \sum_{n=1}^{N} A_{\mathrm{II}n} g_{\mathrm{II}n}(r_k,\theta_k) \right\} \cos\theta_k - T_x \sin\theta_k + T_y \cos\theta_k + R r_k \end{aligned} \tag{8}$$

The $T_x$ and $T_y$ express the rigid-body translation in the $x$ and $y$ directions, respectively, and $R$ is the rigid-body rotation. The subscript $k$ ($k = 1, 2, \ldots$ , M) denotes the index of the function evaluated at a point $(r_k, \theta_k)$ in the displacement field at which the displacement values are $u_{rk}$ and $u_{\theta k}$. The M is the total number of displacement data points used in solving displacement equations. The unknown coordinates around a crack tip may thus be given in polar coordinates as Equation (9):

$$r_k = \sqrt{(x_k - x_0)^2 + (y_k - y_0)^2}, \quad \theta_k = \tan^{-1}\left( \frac{y_k - y_0}{x_k - x_0} \right) \tag{9}$$

The $x_0$ and $y_0$ are the location of the crack tip relative to an arbitrary Cartesian coordinate system. At any point in the displacement field, the coordinates $r_k$ and $\theta_k$, and displacements $u_{rk}$ or $u_{\theta k}$ can be substituted into Equations (7) and (8). Therefore, stress intensity factors, higher-order terms and crack tip locations can be derived automatically from the displacement field determined by the DIC method.

*2.3. Acoustic Emission Event Localization Technique*

The AE technique has been extensively applied for the condition assessment and damage detection for concrete structures, as it was described by Grosse and Ohtsu in their book [13]. In particular, the AE technique can detect crack propagation that occurs not only on the surface but also deep inside the material. Therefore, a large number of AE analyses have been performed on concrete and concrete structures in Golaski's work [24]. One of the most important features of AE technique is the ability to localize the source of an AE event. Through the AE technology and signal localization method, the fracture process in concrete can be observed throughout the loading history. Signal localization is the basis of all analysis techniques used in AE technique. According to the 3-D localization problem, it is exactly determined when four or more travel times are available to calculate the three coordinates and the source time of an event. A least absolute deviation method is proposed as the AE location method in this presentation, and the objective function is defined as Equations (10) and (11)

$$F = \sum_{i=1}^{n} |C_i - C_m| \tag{10}$$

$$C_i = t_i - \left( \sqrt{(x_i - x_0)^2 + (y_i - y_0)^2 + (z_i - z_0)^2} / v \right) \tag{11}$$

The $C_i$ is the source time to the sensor; and $C_m$ is the median of $C_i (i = 1, 2, \ldots, n)$, which can be treated as the real source time. $C_i - C_m$ is the measurement error between calculated source time and real source time. The $t_i$ is the onset time of the sensor, $(x_i, y_i, z_i)$ is the coordinate of the sensor location, and $(x_0, y_0, z_0)$ is the coordinate of the event source location.

### 2.4. Specimen, Loading Condition, and Measurement System

The concrete specimens are prepared with a standard P.O 32.5 Portland cement, which is crushed stone with a maximum diameter of 6 mm and river sand used as coarse and fine aggregate, respectively. The mix proportions are listed in Table 1. The modulus of elasticity and Poisson's ratio are measured through standard material test methods, the specimen size is $100 \times 50 \times 50$ mm$^3$. The measured values of the modulus of elasticity and the Poisson's ratio of the concrete material are $E = 35$ GP$_a$, $\mu = 0.26$, and the compressive strength is 27.5 MP$_a$.

**Table 1.** Concrete mix proportions.

| Water/Cement Ratio | Cement (kg/m$^3$) | Sand (kg/m$^3$) | Aggregate (kg/m$^3$) | Water (kg/m$^3$) |
|:---:|:---:|:---:|:---:|:---:|
| 0.48 | 446 | 593 | 1102 | 214 |

Six single-edge notched concrete beams under the three point bending are used for mode I fracture testing in the present study. Figure 3 shows a schematic representation of the concrete specimen and the locations of the three loading points of a three point bending test. The dimension of the SEN concrete beam is $210 \times 70 \times 23$ mm$^3$, and the span is 170 mm. A notch is made at the center of the concrete specimen edge using a diamond saw of 0.3 mm in thickness, with a length of $a_0 = 10$ mm. In implementing the DIC method, the interest regions of specimen surface are painted with white ink, and then covered with a black dot pattern with spray painting. An example of specimen prepared in this manner is shown in Figure 4.

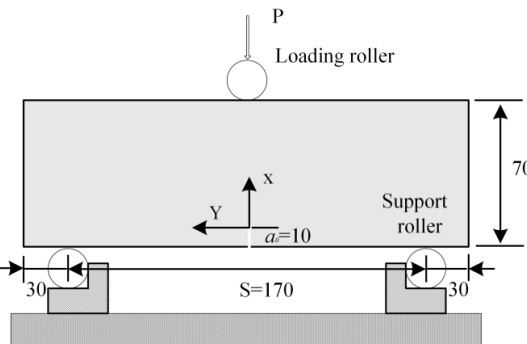

**Figure 3.** Concrete specimen and three-points bending conditions.

A servo hydraulic test machine with a capacity of 50 kN is applied for fracture tests, the displacement rate of the loading plane is set to 0.02 mm/min. In order to ensure contact between the loading system and specimens, a 50 N preload is applied before the testing. During the loading process, the painted area around the crack tip is recorded by a monochromatic charge-coupled device camera (Basler 404k, 2352 × 1720 pixels, BASLER, Ahrensburg, Germany) with a 105 mm focal lens. The specimen failure and crack propagation process is recorded by the camera in a rate of 15 frame pictures per second, then the pictures are stored in a computer automatically. The resolution of all captured images is 0.055 mm/pixel. As shown in Figure 4, sixteen AE sensors with a resonant frequency of approximately 150 kHz are attached to the specimen surface. Six sensors are in front surface and behind surface respectively, and four sensors are on the up and down edge of the specimen respectively.

AE signals are amplified 40 dB gain by a pre-amplifier. The sixteen sensor's signal data in 16-bit is recorded continuously and simultaneously in a frequency of 3 MHz.

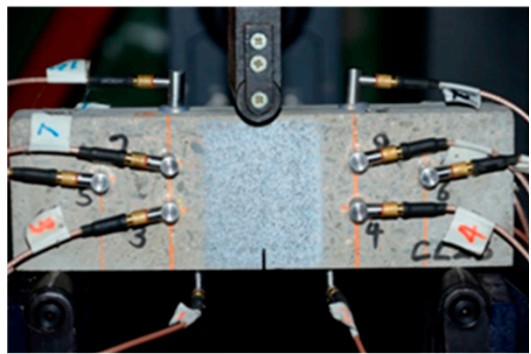

**Figure 4.** Single-edge notched concrete specimen sprayed with a dot pattern.

## 3. Results and Discussion

### 3.1. FPZ Evolution and SIFs

During a typical concrete fracture testing process, 13,089 images are captured by CCD in 872.6 s. A $55 \times 30$ mm$^2$ interest area covering the crack extension trace is shown in Figure 5, the displacement contour map of the interest area is obtained by the DIC technique. The crack tip position and displacement data around the crack tip are used as the initial value of the solution to the iterative Equation (8), then the SIFs and real crack tip position are obtained. As shown in Figure 5, 230 data points are selected around the crack tip, the data point position is marked by black dots. Although the accurate positions of the pre-crack tip can hardly be precisely ascertained at the outset, it is well known that the failure must start from the upper edge of the notch. Therefore, the center point of the notch's upper edge is set as the crack tip's initial position for the first iterative procedure. Then the obtained value of crack tip position is set as a new crack tip position for the consequent iterative procedures. As it is mentioned above, a series crack tip position can be derived from the displacement field around the crack tip during the crack propagation process. These crack tip positions are in the crack extension trace. These crack tip positions are used to calculate crack extension length $\delta_a$ during the FPZ evolution process. As shown in Figure 6, the crack extension length curve and loading curve are plotted together.

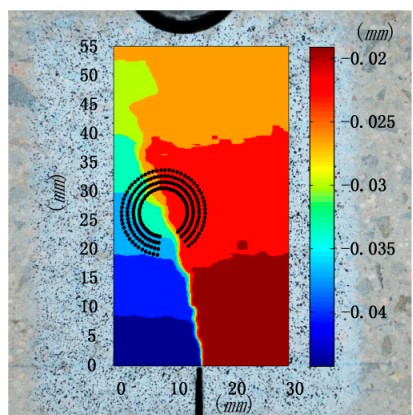

**Figure 5.** The speckle image captured by CCD and the displacement contour map of the interest area.

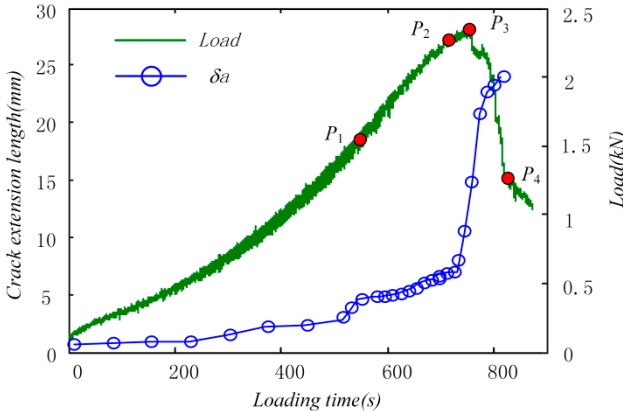

**Figure 6.** The curves of loading and crack extension length.

As shown in Figure 6, the crack extension length gradually increases from $t$ = 577.34 s, the crack extension speed increases sharply after $t$ = 712.26 s, the crack extension speed slows down after $t$ = 752.35 s. Therefore, the first stage is usually called crack stable growth stage or subcritical growth stage. In this stage, micro-cracks are nucleating in front of the pre-crack tip along the notch direction, the crack extension length can be set as FPZ length. As shown in Figure 6, the concrete subcritical growth stage can be identified quantitatively and easily from the crack extension curves, and the FPZ length is about 8.35 mm. The second stage is unstable crack growth stage, micro-cracks coalesce into meso- and macro cracks, the crack extension length increases sharply, and the FPZ length is about 20.5 mm. In the third stage, the crack extension speed is mainly controlled by the test machine displacement rate. So, the concrete FPZ length is obtained from crack extension length curve derived from displacement field by DIC technique proposed in this presentation.

In Figure 7, the mode I SIF $K_I$ curve and loading curve are plotted together. The SIF $K_I$ is expected to be proportional to the applied load $P$ and the square root of the length of preexisting crack $a$, as per the theory of fracture mechanics, while the value of $K_I$ does not increase before $t$ = 577.34 s. The SIF $K_I$ begins to increase from $t$ = 577.34 s, and reaches a critical value $K_I$ = 3.332 MPa·m$^{1/2}$ at $t$ = 752.35 s. Then the curve of $K_I$ drops slightly and tends to a constant value. Hence, the critical value of $K_I$ = 3.332 MPa·m$^{1/2}$ is defined as the mode I fracture toughness $K_{IC}$. As shown in Figure 8, the value of SIF $K_{II}$ are much less than the value of $K_I$. It means that the beam is mainly cracked by tension force. Based on the analysis above, the typical contour maps of horizontal displacement during the loading process are shown in Figure 9.

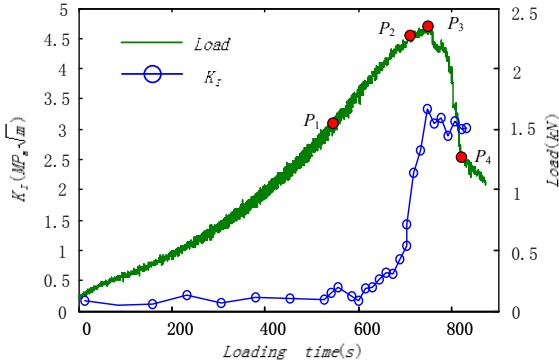

**Figure 7.** The curve of mode-I stress intensity factor $K_I$.

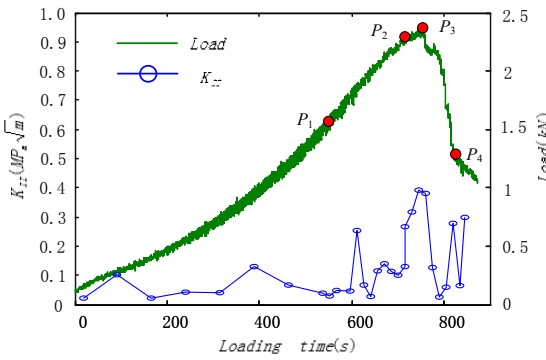

**Figure 8.** The curve of mode-II stress intensity factor $K_{\mathrm{II}}$.

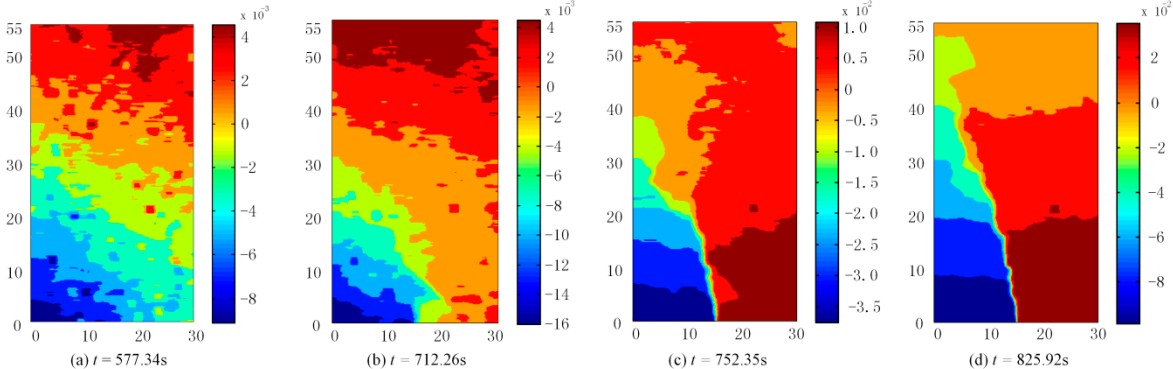

(a) $t$ = 577.34s    (b) $t$ = 712.26s    (c) $t$ = 752.35s    (d) $t$ = 825.92s

**Figure 9.** Horizontal displacement contour maps of the interest area (Unit: mm).

The displacement contour maps are corresponded with four spots marked on loading curve in Figure 6, respectively. Table 2 contained more details on specific values mentioned above, such as the loads, SIFs and crack extension lengths.

**Table 2.** Key experimental result.

| Load Time (t) | Load (kN) | $K_{\mathrm{I}}$ (MPa·m$^{1/2}$) | $K_{\mathrm{II}}$ (MPa·m$^{1/2}$) | $\Delta\alpha$ (mm) | Load/Max Load (%) |
|---|---|---|---|---|---|
| 577.34 | 1.58 | 0.206 | 0.018 | 4.65 | 67.2 |
| 712.26 | 2.29 | 2.545 | 0.281 | 8.35 | 97.4 |
| 752.35 | 2.35 | 3.332 | 0.390 | 15.02 | 100 |
| 825.92 | 1.24 | 3.0065 | 0.281 | 24.63 | 52.8 |

### 3.2. Internal AE Event of FPZ

Figure 10 shows the positions of 16 AE sensors and the last location distribution of the AE event. The 3rd to 8th AE sensors are attached onto the front specimen surface, and the 9th to 14th sensors are attached onto the back specimen surface. The first, second and the 15th and 16th AE sensor are attached onto the downside and upside of specimen edge respectively. A sufficient number of sensors can effectively suppress the influence on the events localization caused by the individual signals attenuation and sensors position. The definition of AE events is defined by a threshold of 100 millivolts. From Figure 10a, the pencil lead break position is near the 9th AE sensor. The total 9035 AE events are checked out from the signal waves. A total of 175 AE event positions are located effectively by the proposed method. It can be found that the location distribution of AE event in Figure 10b agreed with the deformation localization positions in Figure 9. Therefore, the location of AE events can be applied to study the micro-cracking events and the evolution of FPZ internal the concrete specimen.

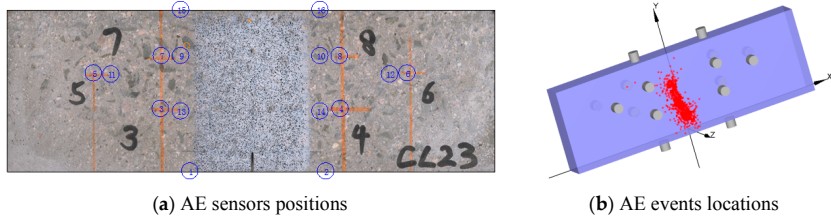

(**a**) AE sensors positions        (**b**) AE events locations

**Figure 10.** The AE sensor position and the AE event location.

From Figure 11, the AE events are marked during the loading process. The marked positions on loading curve are the same as what have been mentioned in the Section 3.1. Therefore, it can be analyzed correspondingly for the internal micro-cracking characteristics and the surface deformation localizations of FPZ. The first AE event is caused by the pencil lead break. The following AE event is caused by the micro-cracking during the FPZ evolution process.

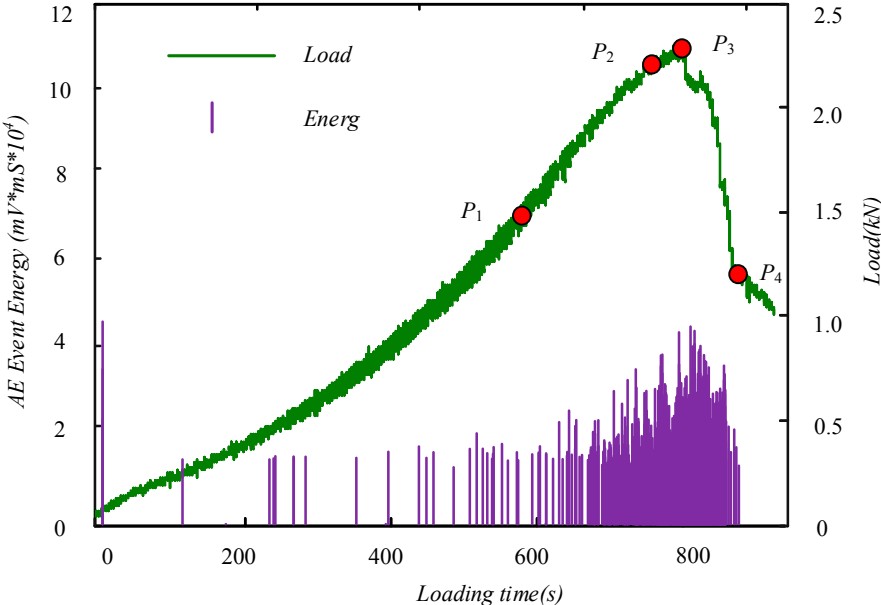

**Figure 11.** The curves of loading and AE event energy distribution.

It can be seen that the most AE events are concentrated between the main crack initial and unstable extension period. AE energy is sharply increased as macro crack forms in a FPZ. The emission energy of the event around the peak loading is higher than the other event. The AE energy in this analysis is defined as the product of amplitude and duration of the emission.

The AE event distribution in the FPZ at loading points $P_1$, $P_2$, $P_3$, and $P_4$ are shown in Figure 12. As it is shown in Figure 12a, the first AE event location is near the up loading roller, but the event is an isolated one. Conversely, the AE events which occurred from the pre-crack tip are a series of event in Figure 12b–d. Meanwhile, these events are distributed in a narrow band, the width of the band is about 10 mm. The band is in accordance with the displacement localization band in Figure 9. Therefore, it can be concluded that the internal AE event band is in accordance with the FPZ ahead of the pre-crack tip. The length of FPZ in Figure 12c,d are in agreement with the crack length shown in Figure 6 observed by the DIC method.

Corresponding to Figure 12, the surface horizontal strain fields are shown in Figure 13. The surface strain is obviously localized in a narrow band in Figure 13c,d. From Figures 12 and 13, it is observed that the FPZ corresponds to both the surface strain localization zone and the internal AE event zone.

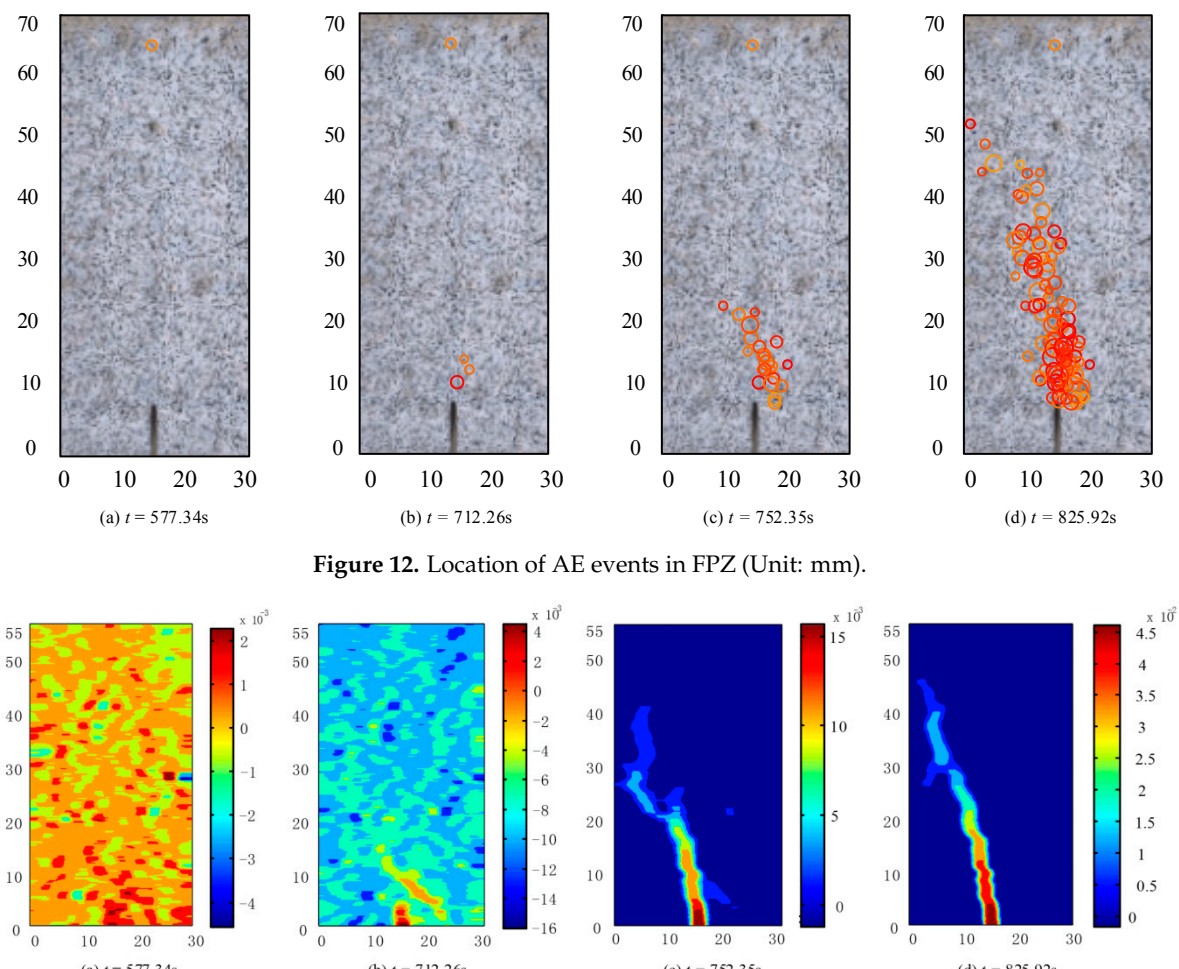

**Figure 12.** Location of AE events in FPZ (Unit: mm).

**Figure 13.** Strain localization ahead the crack tip (Unit: mm).

### 3.3. Concrete FPZ Characteristics

The internal and external characteristics of FPZ in concrete are identified by DIC and AE techniques quantitatively during the mode I fracture extension process. Although the measurement of FPZ length has attracted the attention of most researchers, the quantitative measurement of FPZ length has not yet been realized until now. The crack tip position and FPZ length is measured automatically and quantitatively by the DIC technique proposed in this presentation. The length of FPZ obtained by DIC and AE technique are in good agreement. The maximum of FPZ length is about 20 mm, the experimental results agreed with the hypothesis that the length of FPZ is about three times the maximum aggregate size [24].

The internal micro-cracking process is monitored by AE technique. The cracking source of AE events are located precisely by the proposed location methods. The results show that the internal AE event propagation process is completely consistent with the displacement and strain localization process obtained by DIC method on the surface of concrete specimen. During the concrete crack extension process, the FPZ corresponds to the surface deformation localization band and the internal AE event band. The width of fracture process zone is 3 mm on the surface of specimen and 10 mm in the interior of specimen. It means the internal influence area of crack in concrete is larger than that of FPZ observed on concrete surface.

The measurement of stress intensity factors at crack tip is usually a difficult problem during the FPZ extension process. The SIFs are derived from the displacement field around the FPZ tip by the DIC technique proposed in this presentation. The stress intensity factors indicate the distribution characteristics of the stress field around the FPZ. It should be mentioned that the emission energy

monitored by AE sensor is a relative parameter, which can't be directly applied to quantify the real released energy of AE event. The comparative analysis of AE results based on deformation field and SIFs will help to explain and analyze the experimental results of AE in the FPZ of concrete.

## 4. Conclusions

The following conclusions can be drawn from experiments with Single-edge notched concrete beams under three-points bending using the digital image correlation and acoustic emission technique:

(1) The displacement fields, strain fields, crack tip position, crack extension length, and SIFs are obtained automatically and quantitatively by DIC technique proposed in this presentation, the FPZ is identified by the surface deformation of specimen during the crack extension process.

(2) The internal micro-cracking events of FPZ are localized automatically and dynamically by the proposed AE event location method. The distribution of internal acoustic emission events corresponds to the FPZ on the surface of the specimen.

(3) The length of the FPZ is obtained by DIC and AE technique, the results are consistent with each other.

(4) The internal and surface characteristics of FPZ evolution are identical during the concrete crack extension process. The comparative analysis based on DIC and AE technique is helpful to comprehend the characteristics of concrete FPZ.

**Author Contributions:** S.D. and X.L. conducted the experiments, wrote the paper. K.N. offered useful suggestions for the preparation and writing the paper.

**Funding:** This research was funded by the National Natural Science Foundation of China (41002075), the Open Research Fund Program of State Key Laboratory of Hydroscience and Engineering of Tsinghua University (sklhse-2015-C-06), and the Open Projects of Research Center of Coal Resources Safe Mining and Clean Utilization Liaoning (LNTU17KF15).

**Conflicts of Interest:** The authors declare that they have no conflicts of interest.

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
