# Peer review of "Experimental Study on the Fracture Process Zone Characteristics in Concrete Utilizing DIC and AE Methods"

_applsci, doi:10.3390/app9071346_

Reviewer 1 Report

The authors present a study that correlates the DIC and AE results in an attempt to track the fracture process on concrete samples. The study clearly introduces the topic, discusses limited literature from the field and extensively presents DIC and AE principles. In general, the paper is limited in originality. There are plenty of studies in literature that track the fracture process on concrete using AE and DIC. 

The analysis and results sections should be improved.

It is still not clear how the crack tip is detected using DIC. Are you manually track the point where the crack ends at the deformation maps? Is there a calculating algorithm behind?

This is the most essential information and is not discussed in detail.

Figure 10b is not clear at all and should be improved and focus on the 3D illustration of events population.

Figure 11 can be improved by using AE energy cumulative graph. This way, the energy accumulation can be effectively tracked

As a reviewer, I do have a major objection on the study methodology:

The authors state that the FPZ standing ahead of the crack tip can be seen in Figure 12 (Line 257). This information is correlated to Figure 6 where the crack tip is tracked using AE.

I do not see why the AE events do localize the FPZ and the crack tip position.

AE can indicate the zone where micro-cracking is built surrounding the crack-tip. That is well established knowledge in literature. AE can be indicative of FPZ, but there is no proof that what is measured as AE event distribution corresponds to FPZ. This is big logical jump that triggers many questions and asks for further scientific proof.

Furthermore, did the authors consider the effect of attenuation and sensors position? What if the sensors were placed further, therefore the events localization becomes less sensitive to damage occurrence.

My second remark deals with Figure 13. The authors state that Strain maps shows the FPZ and they go even a step further by quantifying the size of this FPZ based on strain maps analyses.

Be careful! The strain maps of DIC represent an information relative to the color map sensitivity. In other words, if the color map is changed, totally different width of FPZ will be detected.

Indicatively, the color map range changes from Figure 13c to Figure 13d.

The authors should consider fixed color map range (in concrete a typical range is from 0 to 2% of horizontal strain). Additionally, the comment that the FPZ is quantified should be removed as misleading.

Minor typo errors:

Line 214: reachs

Line 247: rephrase, sentence not clearly set.

Author Response

Dear editors and reviewers,

We would like to thank you for spending time on our manuscript. Your comments and suggestions are very valuable. We have considered the comments carefully, and necessary changes have been incorporated into the revised manuscripts as appropriate. In this reply, we will address the comments raised by reviewer #1.  The followings are our replies and summary of changes made.

Response to Reviewer 1 Comments

Point 1: The analysis and results sections should be improved.

Response 1: Thanks for your positive comments! The analysis and results sections have been revised and improved carefully.

Point 2: It is still not clear how the crack tip is detected using DIC. Are you manually track the point where the crack ends at the deformation maps? Is there a calculating algorithm behind? This is the most essential information and is not discussed in detail.

Response 2: The crack tip position and displacement data around the crack tip are used as the initial value of the solution to the iterative Equation (8), then the real crack tip position are obtained. During a typical concrete fracture testing process, 13089 images are captured by CCD in 872.6 seconds. A 55×30 mm2 interest area covering the crack extension trace is shown in Figure 1, the displacement contour map of the interest area is obtained by DIC technique. The specific steps for detecting the crack tip position are as follows:

Figure 1. The speckle image captured by CCD and                           Figure 2. The calculation flow diagram

the displacement contour map of the interest area

First step: The centre point of the precracked notch is set as the initial crack tip position for the first iterative procedure.

Second step: As shown in Figure 1, 230 data points are selected around the crack tip, the data point position is marked by black dots.

Third step: Solve the Equation (8), and obtain the crack tip position.

Fourth step: The obtained value of crack tip position is set as a new initial crack tip position, 230 data points are selected around this crack tip position in the next deformed displacement field. The corresponding displacement datas and the new initial crack tip position are used to Solve the Equation (8) again.

As it is mentioned above, the algorithm is programmed by MATLAB. The calculation flow diagram is shown in Figure 2. 13088 crack tip positions are automatically derived from the displacement field during the crack propagation process.

Point 3: Figure 10b is not clear at all and should be improved and focus on the 3D illustration of events population.

Response 3: Thanks for your positive comments! Figure 10B has been replaced by a high resolution images.

Point 4: Figure 11 can be improved by using AE energy cumulative graph. This way, the energy accumulation can be effectively tracked.

Response 4: Thank you very much! We hope to show the macro crack propagation process. Therefore, the time and energy of each AE event are given in the Figure 11.

Point 5: The authors state that the FPZ standing ahead of the crack tip can be seen in Figure 12 (Line 257). This information is correlated to Figure 6 where the crack tip is tracked using AE. I do not see why the AE events do localize the FPZ and the crack tip position.

Response 5: The surface crack extension length is observed by DIC method, and it is shown in Figure 6. The internal FPZ is observed by AE method, which is shown in Figure 12. The crack tip can not be accurately localized by AE technique. Thus we have corrected this misrepresentation in the revised manuscript.

Point 6: AE can indicate the zone where micro-cracking is built surrounding the crack-tip. That is well established knowledge in literature. AE can be indicative of FPZ, but there is no proof that what is measured as AE event distribution corresponds to FPZ. This is big logical jump that triggers many questions and asks for further scientific proof.

Response 6: This is really a good question. In this report, we find that the distribution characteristics of internal AE events are consistent with the characteristics of FPZ observed by DIC technique. Therefore, we think the AE event distribution corresponds to FPZ.

Point7: Furthermore, did the authors consider the effect of attenuation and sensors position? What if the sensors were placed further, therefore the events localization becomes less sensitive to damage occurrence.

Response 7: Sixteen AE sensors were placed on the surface of the concrete specimen. A sufficient number of sensors can effectively suppress the influence on the events localization caused by the individual signals attenuation and sensors position. 

Point 8: My second remark deals with Figure 13. The authors state that Strain maps shows the FPZ and they go even a step further by quantifying the size of this FPZ based on strain maps analyses. Be careful! The strain maps of DIC represent an information relative to the color map sensitivity. In other words, if the color map is changed, totally different width of FPZ will be detected. Indicatively, the color map range changes from Figure 13c to Figure 13d.The authors should consider fixed color map range (in concrete a typical range is from 0 to 2% of horizontal strain). Additionally, the comment that the FPZ is quantified should be removed as misleading.

Response 8: In our early studies, we tried to calculate the length and width of the FPZ by setting a strain threshold, but the results are not credible. The results can only be used for preliminary reference. This is also the reason why we propose a new method to locate the crack tip. Therefore, the comment that the FPZ is quantified based on strain maps has been removed in the revised manuscript.

Point 9: Minor typo errors: Line 214: reachs; Line 247: rephrase, sentence not clearly set.

Response 9: The reviewer’s detailed checking on the typos and grammatical errors in the manuscript is highly appreciated. The manuscript is thoroughly checked, and the typos and grammatical errors are corrected accordingly. The sentence in 247 line has been rewritten.

Reviewer 2 Report

The Authors presented experimental study on the fracture process zone in concrete element with DIC and AE method.

The subject is interesting for the readers of the papers.

The paper should be improved:

English flaw: The presentation is not always clear.

Section 2 reports well known information. It can be shortened.

An estimation of the size of the process zone would be interesting and if it remains constant within the crack propagation.

A spectral analysis of the acoustic emission would be interesting.

Author Response

Dear editors and reviewers,

We would like to thank you for spending time on our manuscript. Your comments and suggestions are very valuable. We have considered the comments carefully, and necessary changes have been incorporated into the revised manuscripts as appropriate. In this reply, we will address the comments raised by reviewer #2.  The followings are our replies and summary of changes made.

Response to Reviewer 2 Comments

Point 1: English flaw: The presentation is not always clear.

Response 1: Thanks very much! The presentation is thoroughly checked, and the typos and grammatical errors are corrected accordingly.

Point 2: Section 2 reports well known information. It can be shortened.

Response 2: Thanks very much! Section 2 has been shortened in the revised manuscript.

Point 3: An estimation of the size of the process zone would be interesting and if it remains constant within the crack propagation.

Response 3: This is really a good question. We want to confirm whether there is a constant value of FPZ by increasing the size of the concrete specimen. If we can find this constant value, it can be set as the critical value of FPZ. Then this work will be more valuable. We will continue to carry out this work in the future.

Point 4: A spectral analysis of the acoustic emission would be interesting.

Response 4: Thanks for your positive comments! We haven't completed the spectral analysis algorithm yet. In the following research work, we should carry out the spectral analysis of acoustic emission.

Round  2

Reviewer 1 Report

-

The paper quality has been partially improved following the first revision round. 

However, the authors did not manage to comprehensively respond to the review questions and according to me the manuscript is still lacking in quality. 

There were two fundamental questions pointed out and the responses were not satisfying. Indicatively, the authors insist to define fracture process zone based on AE events dudtribution and DIC strain maps. For me these are fundamental errors and additional discussion is needed to defend these points.

I recommend the acceptance of the paper only if the other reviewers see a clear originality and inteeein in this study.

Author Response

The authors highly appreciate the comments of the reviewer. As per the comments of the reviewer, following please find our responses.

The first fundamental questions and response:

Point 1: The authors state that the FPZ standing ahead of the crack tip can be seen in Figure 12 (Line 257). This information is correlated to Figure 6 where the crack tip is tracked using AE. I do not see why the AE events do localize the FPZ and the crack tip position.

Response 1: In Figure 6, the crack tip and crack extension length are identified by DIC technique, not by AE technique. The crack tip can not be identified by AE technique. We have corrected this misrepresentation in the revised manuscript. The FPZ is defined as a micro-cracking region ahead of the pre-cracked tip, micro-cracking can cause AE events. The internal distribution of AE events can be used to localize FPZ. Zietlow and Muralidhara respectively measured intrinsic process zone and FPZ size by the AE event distribution in their research work.

1.        Zietlow, W.K.;  Labuz J.F. Measurement of the intrinsic process zone in rock using acoustic emission. International Journal of Rock Mechanics & Mining Sciences 1998, vol.35, no.3, pp. 291-299.

2.        Muralidhara, S.; Prasad, B.K.R.; Eskandari, H. Fracture process zone size and true fracture energy of concrete using acoustic emission. Construction & Building Materials 2010, vol.24, no.4, pp. 479-486.

Point 2: AE can indicate the zone where micro-cracking is built surrounding the crack-tip. That is well established knowledge in literature. AE can be indicative of FPZ, but there is no proof that what is measured as AE event distribution corresponds to FPZ. This is big logical jump that triggers many questions and asks for further scientific proof.

Response 2: In our study, the distribution of internal acoustic emission events is consistent with the length and shape of FPZ identified by DIC technique. Therefore, the AE event distribution is used to indentify the internal FPZ.

Point 3: Furthermore, did the authors consider the effect of attenuation and sensors position? What if the sensors were placed further, therefore the events localization becomes less sensitive to damage occurrence.

Response 3: Sixteen AE sensors were attached around the pre-cracked tip in the testing. As long as four of the 16 sensors can obtain effective signals, AE events can be localized. Therefore, the localization result will not be affected by the position of a few sensors.

The second fundamental questions and response:

Point 1: My second remark deals with Figure 13. The authors state that Strain maps shows the FPZ and they go even a step further by quantifying the size of this FPZ based on strain maps analyses. Be careful! The strain maps of DIC represent an information relative to the color map sensitivity. In other words, if the color map is changed, totally different width of FPZ will be detected. Indicatively, the color map range changes from Figure 13c to Figure 13d.The authors should consider fixed color map range (in concrete a typical range is from 0 to 2% of horizontal strain). Additionally, the comment that the FPZ is quantified should be removed as misleading.

Response 1: This is really a good question. It is indeed inaccurate to use the width of strain localization zone to determine the width of FPZ. Therefore, the content of quantitative FPZ width based on strain map is deleted in the revised manuscript.

Furthermore, English language of this paper has been improved by Kumar Nawnit, one of the authors, who is a native speaker of English.

Reviewer 2 Report

The revised version of the manuscript can be accepted for pubblication.

Author Response

Thanks for your comments.

English language has been improved by the Kumar Nawnit, one of the authors of this paper, who is a native speaker of English.

Round  3

Reviewer 1 Report

-I have already submitted my revision on the paper. Please see the second round of revisions as submitted two weeks ago.

Round  4

Reviewer 1 Report

Following the comments of the authors at this second revision, I agree to accept the publication of this manuscript.